# Combinations of genes at the 16p11.2 and 22q11.2 CNVs contribute to neurobehavioral traits

**Mikhail Vysotskiy**[1,2,3], **Autism Working Group of the Psychiatric Genomics Consortium**[¶], **Bipolar Disorder Working Group of the Psychiatric Genomics Consortium**[¶], **Schizophrenia Working Group of the Psychiatric Genomics Consortium**[¶], **Lauren A. Weiss**[1,2,3]*

**1** Institute for Human Genetics, University of California San Francisco, San Francisco, California, United States of America, **2** Department of Psychiatry and Behavioral Sciences, University of California San Francisco, San Francisco, California, United States of America, **3** Weill Institute for Neurosciences, University of California San Francisco, San Francisco, California, United States of America

¶ Membership of the Psychiatric Genomics Consortium are listed in the S1 Note.
* lauren.weiss@ucsf.edu

## Abstract

The 16p11.2 and 22q11.2 copy number variants (CNVs) are associated with neurobehavioral traits including autism spectrum disorder (ASD), schizophrenia, bipolar disorder, obesity, and intellectual disability. Identifying specific genes contributing to each disorder and dissecting the architecture of CNV-trait association has been difficult, inspiring hypotheses of more complex models, such as multiple genes acting together. Using multi-tissue data from the GTEx consortium, we generated pairwise expression imputation models for CNV genes and then applied these elastic net models to GWAS for: ASD, bipolar disorder, schizophrenia, BMI (obesity), and IQ (intellectual disability). We compared the variance in these five traits explained by gene pairs with the variance explained by single genes and by traditional interaction models. We also modeled polygene region-wide effects using summed predicted expression ranks across many genes to create a regionwide score. We found that in all CNV-trait pairs except for bipolar disorder at 22q11.2, pairwise effects explain more variance than single genes. Pairwise model superiority was specific to the CNV region for all 16p11.2 traits and ASD at 22q11.2. We identified novel individual genes over-represented in top pairs that did not show single-gene signal. We also found that BMI and IQ have significant regionwide association with both CNV regions. Overall, we observe that genetic architecture differs by trait and region, but 9/10 CNV-trait combinations demonstrate evidence for multigene contribution, and for most of these, the importance of combinatorial models appears unique to CNV regions. Our results suggest that mechanistic insights for CNV pathology may require combinational models.

## Author summary

Copy number variants (CNVs) at 16p11.2 and 22q11.2 are associated with neurobehavioral traits including ASD, bipolar disorder, schizophrenia, BMI, and IQ). Previously, we

**Data Availability Statement:** Individual-level genotypes for Psychiatric Genomics Consortium cohorts can be obtained by applying at https://pgc.unc.edu/for-researchers/data-access-committee/data-access-information/ Summary level data from the PGC is at https://pgc.unc.edu/for-researchers/download-results/. Summary-level genetic datasets for BMI and IQ are available to freely download from GIANT BMI (https://portals.broadinstitute.org/collaboration/giant/index.php/GIANT_consortium) and CNCR IQ (https://ctg.cncr.nl/software/summary_statistics). Individual-level UK Biobank data can be obtained by application at https://www.ukbiobank.ac.uk/enable-your-research/apply-for-access PrediXcan single-gene genome-wide models are available to download at predictdb.org. GTEx genotypes and phenotypes are requestable on dbGAP (https://www.ncbi.nlm.nih.gov/projects/gap/cgi-bin/study.cgi?study_id=phs000424.v8.p2). Summary statistics from association studies performed in this article are located in S5, S6 and S7 Tables.

**Funding:** This work was supported by National Institute of Mental Health R01 MH107467 to LAW. The funding body had no role in study design, data collection and analysis, decision to publish, or preparation of the manuscript.

**Competing interests:** The authors have declared that no competing interests exist.

attempted to identify individual genes within these CNVs relevant for each trait, but found that many CNV-trait pairs did not demonstrate single-gene association. Here, we use similar methodology to assess whether the effect of CNV genes on the same traits could be better explained by pairs of genes acting together. We found that in nearly all cases, pairs of genes explained trait variance better than single genes. In several cases, specific genes contributed to traits disproportionately in pairs, but not individually. Additionally, we tested for region-wide association using all genes in the region, and found that both the 16p11.2 and 22q11.2 regions had a significant effect on BMI and IQ. Our results demonstrate that the genetic architecture of CNV-trait associations is multigenic and may vary across CNVs and traits.

## Introduction

Copy number variants (CNVs) at 16p11.2 and 22q11.2 contribute to neurobehavioral disorders including autism spectrum disorder (ASD), schizophrenia, bipolar disorder, intellectual disability, and obesity [1–11]. Specific gene-trait contributions at these regions have proven difficult to find. Single-gene fine-mapping approaches have been challenging due to a lack of highly-penetrant point mutations in these genes and inconsistent findings in animal models [12–15]. A potential reason for the lack of clear gene-phenotype relationships is that the architecture may be more complicated than single-gene contributions to each trait [16]. More complex models are good candidates for *in silico* analysis, as multiple hypotheses can be efficiently assessed in parallel.

Data in humans and mice suggest that the expression of 16p11.2 and 22q11.2 CNV genes is consistently upregulated/downregulated in duplication/deletion carriers [17–20]. From this observation, we can propose that gene expression dysregulation (and potential downstream protein expression) is likely to be a pathophysiological mechanism of CNV-associated traits. This implies that examination of the consequences of gene expression variation for neurobehavioral traits may indirectly uncover the genetic architecture of CNV-phenotype association. However, gene expression data for cases affected with neurobehavioral traits remains limited in availability and ambiguous with respect to causality. Instead, we can use expression-imputation methodology and rely on genetic data, available for a far greater number of (control) individuals, to predict gene expression under the assumption that genetic regulation is similar in cases and controls. This method allows us to analyze the architecture at a gene level (rather than individual SNPs) and because it is based on germline genetics, is not affected by potential confounding influences on gene expression such as age, chronic illness, medication use, and circumstances of death and tissue preservation. eQTLs (in our case, SNPs used for expression prediction) are less likely to affect genes in a context-dependent manner, as eQTL-linked genes are less likely to be affected by enhancer activity compared to GWAS-linked genes [21]. Given that our regions of interest have trait associations via CNVs but very limited GWAS signal for the same traits, using eQTLs and expression prediction may find additional information missed by GWAS analyses.

Previously, we used expression imputation to test whether individual genes at the 16p11.2 and 22q11.2 CNV regions were contributing to our five traits of interest (ASD, schizophrenia, bipolar disorder, intellectual disability, and obesity) [22]. We found contributions of *INO80E* to schizophrenia and body mass index (BMI) and of *SPN* to BMI and IQ, both at 16p11.2. However, no individual genes were associated with 22q11.2 traits, despite using equally-powered genetic datasets. No genes at 16p11.2 were significantly associated with ASD or bipolar disorder using our experiment-wide threshold. These lack of findings in light of the overall

success of our approach were disappointing given the high prevalence of traits such as ASD in 16p11.2 CNV carriers and schizophrenia in 22q11.2 deletion carriers. One explanation for lack of gene-trait association is that individual genes may not be independent contributors to these traits, rather the genetic architecture is combinatorial. Promisingly, it was found that several pairs of 16p11.2 genes in *Drosophila* showed evidence of stronger effects on eye phenotypes than individual genes, and double mutants of 16p11.2 genes in zebrafish led to hyperactivity and body size phenotypes [15, 23]. Thus, we aimed to investigate combinatorial associations in our traits of interest in humans.

In a CNV carrier, all genes within the breakpoints are duplicated or deleted, typically with a similar increase/decrease of expression across all genes. In our previous study, we considered the level of expression of any individual gene, and its effect on relevant phenotypes in non-carriers. Here, we consider two additional models in non-carriers (Fig 1a). First, as a feasible way to model multigene effects at specific pairs of genes, for each gene pair we look for trait association with expression increases or decreases across two genes. Second, we analyze association patterns when gene expression trends towards being upregulated or downregulated across the whole region as a way to capture effects of more than two genes.

## Results

We predicted the expression of individual CNV genes (using publicly available elastic net models) and pairs of CNV genes (using elastic net models trained on GWAS SNPs) across GTEx tissues. We also selected matched control regions for comparison with the CNV region. First, we identified significant genes and gene pairs through association analysis with five traits (using the control region genes as a null distribution to test for significance). Next, we compared the trait variance explained by single gene models vs pairwise models, as well as the specific genes with top associations in single gene vs pairwise models. Finally, we used a rank scoring approach to create region-wide scores to test for a polygenic contribution of CNV genes across the region. Fig 1b summarizes this analysis design.

### Summary of individual gene results

We have updated our single-gene prioritization from our previous study using new models from GTEx version 8 and new data from schizophrenia PGC wave 3 (Table 1) [24, 25]. With this enhancement, we find one 22q11.2 gene (*PPIL2*) significantly associated with schizophrenia at a permutation-based threshold (Table 1 and S5 Table). We note that the permutation-based threshold ($P <$ median of 5th percentiles of control region *P*-values) is less conservative than the experiment-wide thresholds used in previous analysis [22]. However, we can identify five top genes at 22q11.2 associated with BMI (*YDJC*, *CCDC116*, *PPIL2*, *THAP7*, *UBE2L3*), primarily located outside the canonical CNV region (LCR D-E), three with bipolar disorder (*TMEM191B*, *TUBA8*, *PPIL2*), six with ASD (*CLTCL1*, *AC004471.10*, *UFD1L*, *DGCR14*, *CCDC188*, *DGCR9*), and two with IQ (*SEPT5*, *LINC00896*) (Table 1 and S5 Table). The top genes associated with ASD at 22q11.2 are located in the LCR A-B part of the variant, consistent with a previous study [26]. At 16p11.2, the majority of genes tested (31/38) show an association with BMI. We find that, after updating single-gene prediction models to GTEx v8, *SPN* is no longer a major driver of BMI and IQ, as the best predictive SNPs in the most up-to-date version of GTEx did not overlap with top *SPN* SNPs as before; however, new models for *SULT1A4* indicated this gene as a major contributor to both BMI and IQ (S5 Table). *INO80E* and *KCTD13* remained associated with BMI. We find that *INO80E* is a top association with bipolar disorder and ASD; this gene previously showed suggestive bipolar disorder association but did not meet experiment-wide significance criteria even with the updated models [22].

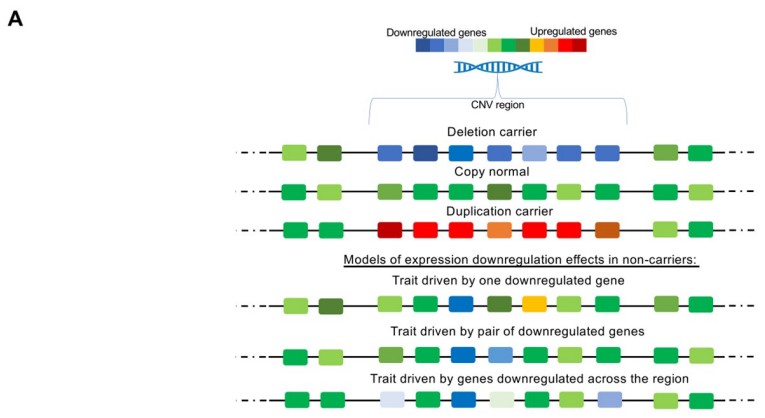

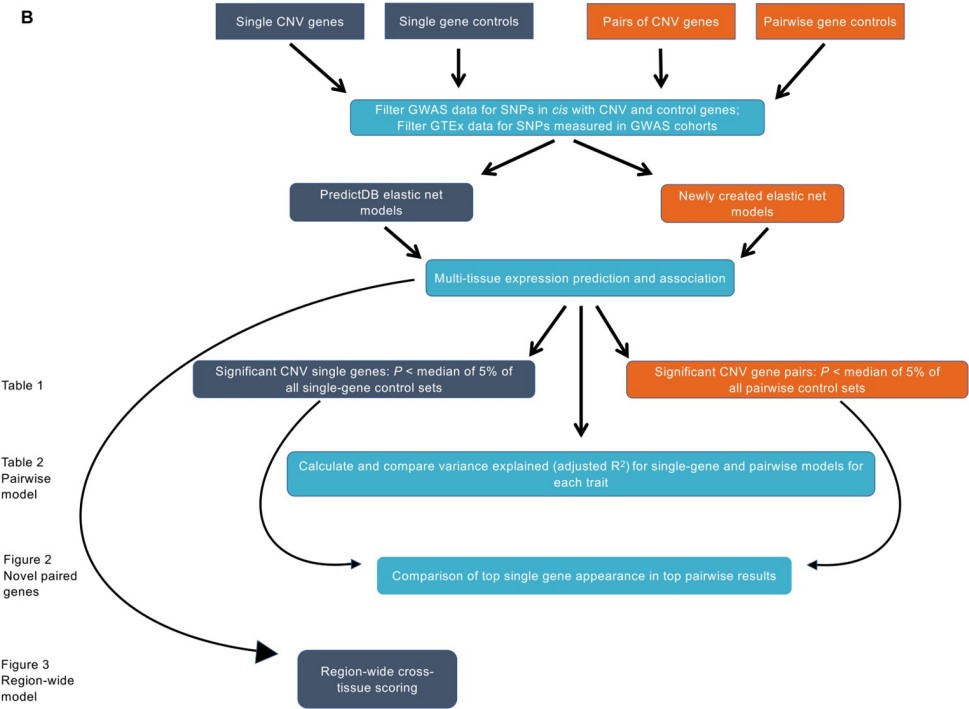

**Fig 1. Models of CNV pathogenicity via gene expression and analysis design.** (A) Rectangles represent individual genes in a chromosomal location. Warmer colors represent increased mRNA expression. Cooler colors represent decreased mRNA expression. Greens represent population average mRNA expression. Top: Within a CNV region, deletion carriers have reduced expression across the majority of genes, duplication carriers have increased expression across the majority of genes, and copy normal individuals have "average" levels of expression across the majority of genes. These increases and decreases are specific to the CNV region experiencing increased or decreased DNA copies (potential positional effects on flanking genes not shown). Bottom: Three models of how gene expression downregulation in a CNV region may influence a CNV-associated trait in non-carriers. In the first model, decreased expression of a single gene is sufficient. In the second model, a trait is impacted when two specific genes both have reduced expression. In the third model, the trait becomes more likely due to reduction of expression in many genes across the region. These three models are utilized in our study. (B) Visual diagram of analysis design.

**Table 1. Proportion of significantly associated ($P$ < median of 5[th] percentiles of control region $P$-values) single genes (single) and pairwise gene sums (pairs) for each trait and CNV.**

| Trait | 16p11.2 | | 22q11.2 | |
|---|---|---|---|---|
| | N single (%) | N pairs (%) | N single (%) | N pairs (%) |
| ASD | 8/42 (19%) | 273/1542 (18%) | 6/65 (9%) | 282/3654 (8%) |
| Bipolar | 5/37 (14%) | 225/1546 (15%) | 3/59 (5%) | 137/3669 (4%) |
| Schizophrenia | 21/37 (57%) | 702/1543 (45%) | 1/59 (2%) | 129/4267 (3%) |
| BMI | 31/38 (82%) | 1212/1554 (78%) | 5/52 (10%) | 176/3229 (5%) |
| IQ | 5/38 (13%) | 80/1545 (5%) | 2/65 (3%) | 33/4052 (1%) |

## Predicting expression of pairs of 16p11.2 and 22q11.2 genes

We trained elastic net models for pairs of 16p11.2 and 22q11.2 genes (both coding and non-coding when possible) using dataset-specific SNP lists to maximize overlap. In general, the model quality (as measured by the performance $R^2$) of pairwise models was in-between that of the two genes that it comprised, as expected (S2 Table). Pairwise predictor SNPs were localized throughout the region rather than immediately adjacent or only between the two individual genes. Pairwise predictors generally did not overlap individual gene predictors, though some were members of the same LD blocks (a representative case is shown in S1 Fig). In addition, we trained pairwise models for genes in control regions (N = 38 regions for 16p11.2 and N = 28 regions for 22q11.2).

## Pairwise association signal is oligogenic

Using our pairwise models to perform association analysis, we found that there were 278 16p11.2 and 282 22q11.2 pairs significantly associated with ASD, 225 16p11.2 and 137 22q11.2 pairs associated with bipolar disorder, 702 16p11.2 and 129 22q11.2 pairs associated with schizophrenia, 80 16p11.2 and 33 22q11.2 pairs associated with IQ and 1,212 16p11.2 and 176 22q11.2 pairs associated with BMI (Table 1, S6 and S7 Tables). Top pairs primarily consisted of two coding genes, consistent with the larger number of coding-coding pairs that had a high-quality prediction model. The proportion was similar for most comparisons, suggesting that testing of correlated pairs does not skew our results. Our results suggest that a single strong association is unlikely to drive our interpretation in most cases. We thus find that pairwise association signal is oligogenic, spread across many pairs rather than enrichment specific to top outlier results (S2 and S3 Figs). Due both to the eQTL sharing between pairwise prediction models as well as to the sharing of genes across pairs, we are unable to use our approach to confidently identify specific candidate gene pairs; several pairs of potential interest are noted in the Discussion section.

## Pairwise prediction models explain more trait variance than single-gene or interaction models

To assess whether analyzing pairs of genes provided more information than individual genes, we calculated how much variance in CNV-associated traits was explained by predicted gene expression as the adjusted $R^2$ of linear models of individual gene expression predictions, pairwise additive gene expression predictions, and two-gene interaction models. We calculated the proportion of tissue-cohort pairs for which pairwise gene expression was the best predictor. In all trait-region pairs, with the exception of bipolar disorder at 22q11.2, we found that the trait variance explained was greater for gene pairs proportionally more often than either single

**Table 2. Counts of the model estimated to explain most trait variance for each tissue-cohort pair.** Best model is bolded in each case. *P*-value represents a chi-square test comparing the proportion of pairwise to non-pairwise counts between CNV regions and controls.

| Region | Trait | CNV Region single/interaction/pairwise (% pairwise) | All Control Regions single/interaction/pairwise (% pairwise) | region-specific P-value |
|--------|-------|------------------------------------------------------|---------------------------------------------------------------|--------------------------|
| 16p11.2 | ASD | 205/169/**243** (39%) | 5891/**7588**/6387 (32%) | 0.00012 |
|  | Bipolar | 359/390/**721** (49%) | 14806/19593/**19631** (36%) | < 2.2x10⁻¹⁶ |
|  | Schizophrenia | 754/730/**1554** (51%) | 26589/37723/**47784** (43%) | < 2.2x10⁻¹⁶ |
|  | BMI | 0/0/**49** (100%) | 48/159/**1744** (89%) | 0.016 |
|  | IQ | 0/0/**49** (100%) | 98/232/**1565** (83%) | 0.0013 |
| 22q11.2 | ASD | 174/196/**267** (42%) | 4909/**7016**/5313 (31%) | 1.3x10⁻⁹ |
|  | Bipolar | **536**/435/499 (34%) | 11642/**15167**/14381 (35%) | 0.44 |
|  | Schizophrenia | 871/816/**1155** (41%) | 19632/28147/**35053** (42%) | 0.07 |
|  | BMI | 0/0/**49** (100%) | 17/68/**1258** (94%) | 0.069 |
|  | IQ | 7/17/**25** (51%) | 20/92/**1041** (90%) | <2.2x10⁻¹⁶ |

genes or interactions (Table 2). To confirm whether this phenomenon was CNV region-specific or a polygenic property of the trait, we additionally performed this analysis for control gene sets. For all traits tested at 16p11.2, the proportion of pairwise models exceeding single or interaction was greater than that of control regions ($P < 0.05$). At 22q11.2, the CNV region performed better than control regions in ASD ($P = 1.3x10^{-9}$), but schizophrenia, bipolar disorder, and IQ showed similar or higher proportion pairwise best performance in control regions (Table 2).

## Patterns of genes most represented in associated pairs differ by phenotype

We wanted to know whether the pairwise associations were primarily comprised of genes with independent association signal or indicated genes with uniquely combinatorial effects. The results were strikingly different across traits (S8 Table). In two cases–bipolar disorder at 16p11.2 and bipolar disorder at 22q11.2 –one gene stood out as a disproportionate contributor to pairs (although in 16p11.2 there was a noncoding gene just on the threshold ~2.5 SD), but in both regions the disproportionate contributor was also a top single gene association (*INO80E* and *PPIL2* respectively). No single gene contributed disproportionally to pairs for ASD, schizophrenia, or BMI at 16p11.2 (S8 Table and S4 Fig). At 22q11.2, however, ASD pairs disproportionately included five genes—*AC004471.10*, *CLTCL1*, and *CCDC118*, which were in the top ASD single genes, as well as *DGCR2* and *DGCR6* which we did not pick up as top single gene associations. For IQ at 22q11.2, *COMT* was a gene that disproportionately appeared in pairs and was not a top single gene (S8 Table and S5 Fig). The remaining genes over-represented in pairs at both 16p11.2 and 22q11.2 were primarily non-coding genes that did not have significant single-gene models, demonstrating potential regulatory effects of non-coding genes on CNV coding genes (S8 Table and S4 and S5 Figs). Three of the common patterns–a top single gene disproportionately represented, no genes disproportionately represented, and novel genes disproportionally represented–are illustrated in Fig 2.

## Region-wide contributions of 16p11.2 and 22q11.2 CNVs to phenotype

After comparing the impacts of single genes and pairs of CNV genes on neurobehavioral traits, we wanted to test combinations greater than pairwise, but feasibility limited our combinatorial testing. Therefore, we considered a polygene region-wide model: whether the average deviation of the multigenic region contributes to a phenotype. We assigned a region-wide score to

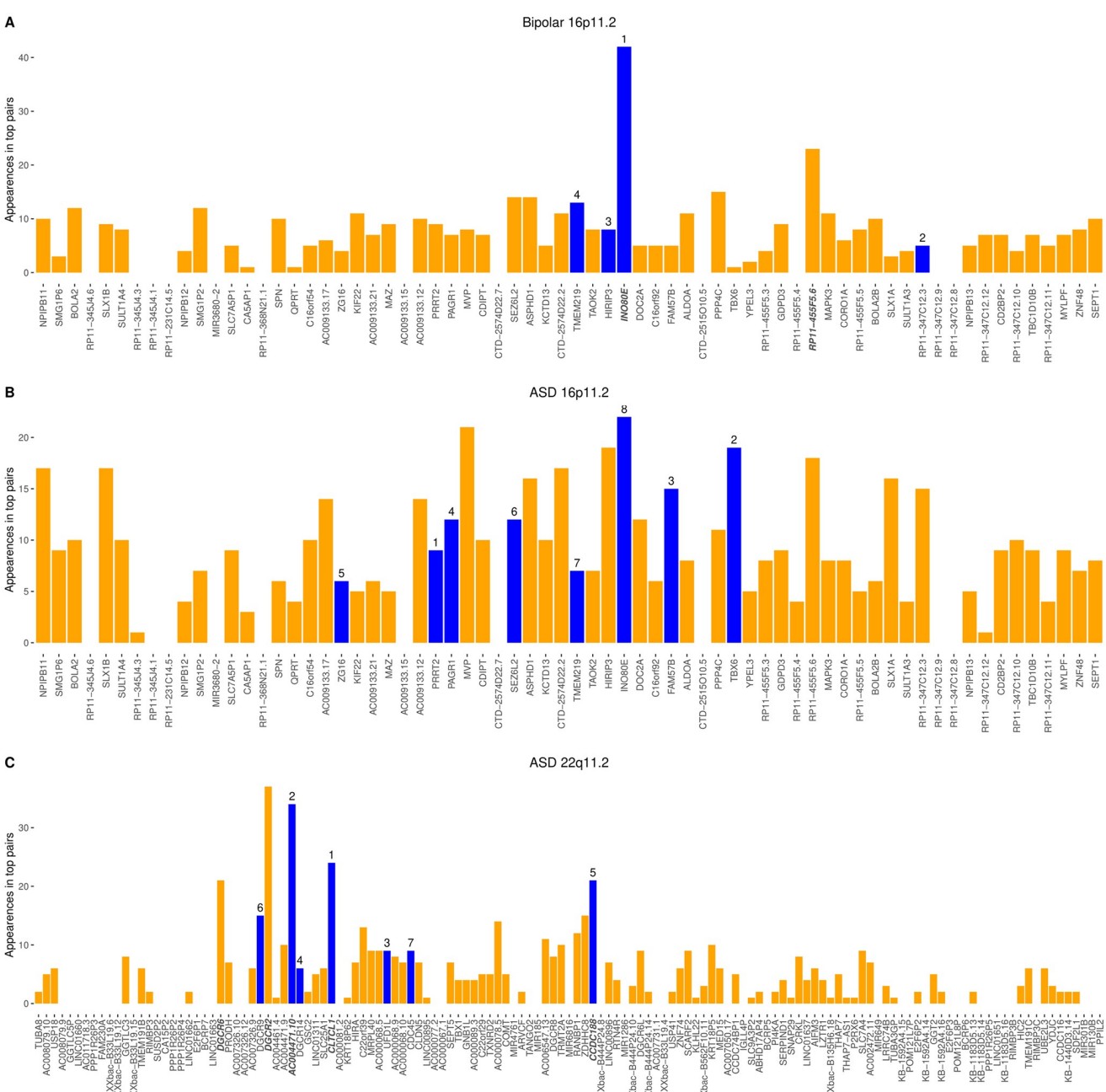

**Fig 2. Three representative examples of the distribution and overrepresentation of individual genes involved in significant pairs.** Y-axis: counts of the number of times each gene is part of a significant pair (permutation $P$-value < median of 5th percentiles of control region $P$-values). Bars in blue represent genes significant (permutation $P$-value < median of 5th percentiles of control region $P$-values) in the single gene model for the same trait, with rank indicated above the bar. Bars in orange represent genes not significant in a single gene model. X-axis: genes in chromosomal order. Disproportionately overrepresented genes (mean + 2.5 standard deviations) are bolded. The counts are tabulated in S8 Table. (A) For bipolar disorder at 16p11.2, one gene, *INO80E* is disproportionately involved in significant pairs; this gene is also a significant single gene. A borderline-overrepresented noncoding gene is also present. (B) For ASD at 16p11.2, no genes are overrepresented in significant pairs. (C) For ASD at 22q11.2, both genes that were significant in single gene analyses (like *CLTCL1*) as well as genes that were not significant on their own (like *DGCR2*) show disproportionate overrepresentation in pairs.

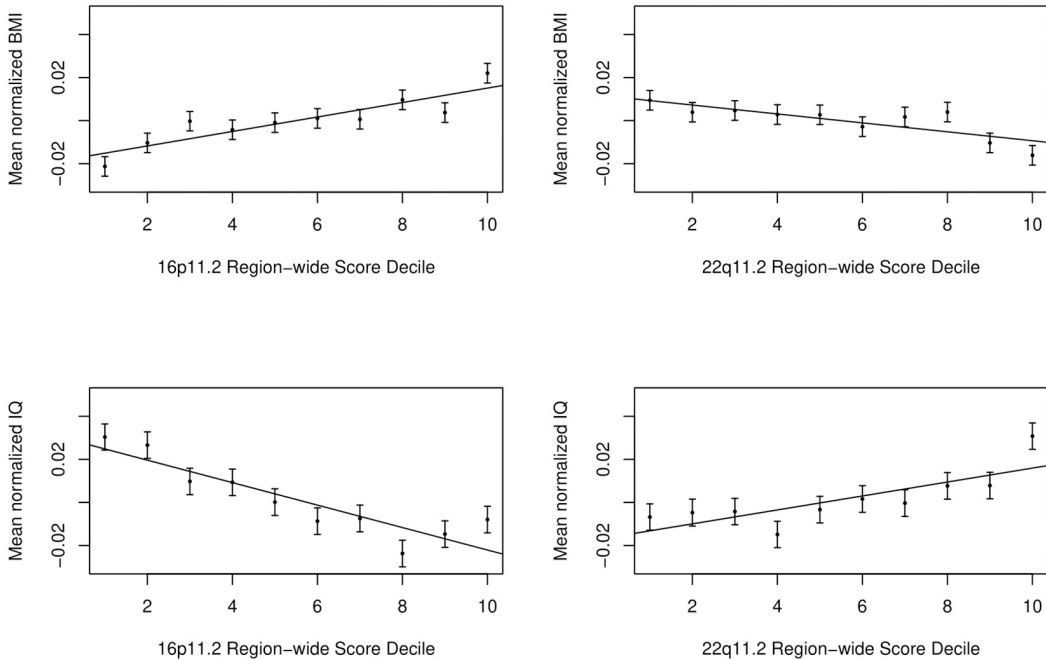

**Fig 3. IQ and BMI phenotypes are associated with a region-wide score.** Region-wide scores were generated by ranking each individual based on their sum predicted CNV gene expression levels (see Methods). Region-wide scores across individuals are shown binned into deciles from lowest to highest predicted expression across the region and the mean (dot) and standard error (bars) of BMI and IQ values for each region-wide score decile are plotted. Best fit line across deciles is shown. Association p-values, based on the entire (not binned) UK Biobank dataset: BMI 16p11.2 $P = 2.0\times10^{-11}$; BMI 22q11.2 $P = 0.0001$; IQ 16p11.2 $P = 8.7\times10^{-15}$; IQ 22q11.2 $P = 1.8\times10^{-6}$.

each individual and tested whether scores were significantly different between cases and controls or correlated with quantitative traits. We found that the region-wide score was positively correlated with BMI for 16p11.2 genes ($P = 2.0\times10^{-11}$) and negatively correlated for 22q11.2 genes ($P = 0.0001$) (Fig 3). IQ was also negatively correlated with region-wide score for 16p11.2 genes ($P = 8.7\times10^{-15}$) and 22q11.2 genes ($P = 1.8\times10^{-6}$) (Fig 3). None of the categorical traits showed a significant region-wide contribution (S6 Fig).

## Discussion

Our study aimed to provide insight into the genetic architecture of the 16p11.2 and 22q11.2 copy number variants. We modeled the neurobehavioral trait consequences of pairs of genes expressed in the same direction, extending our previous single-gene analysis (Fig 4). Both 16p11.2 and 22q11.2 had pairs of genes associated with all tested phenotypes based on a permutation-based threshold, however, despite a larger number of genes tested in 22q11.2, the count of associated genes was larger for 16p11.2 gene pairs. We found that for nearly all traits tested, variance in phenotype was better explained by pairs of genes than by single genes or traditional interaction models. The only exception was bipolar disorder at 22q11.2, where single genes explain more variance. However, for schizophrenia, BMI, and IQ at 22q11.2 the pairwise model was not specific to the CNV regions but appeared to be a trait-based property of genetic architecture extending to matched control regions. These findings suggest that the pairwise effects are different between regions. The advantage of summed pair models in control regions over single and interaction models in 7 of 10 traits–even when it was less pronounced than

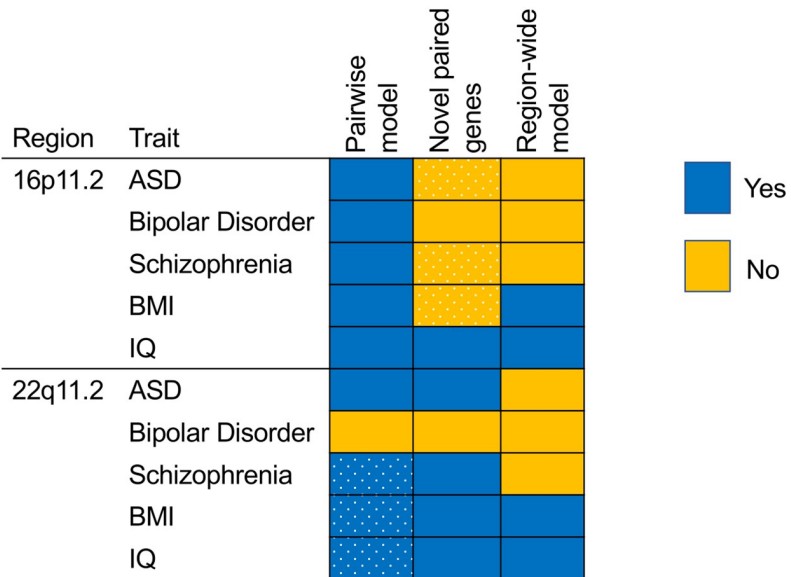

**Fig 4.** Summary of patterns of new information gained from multi-gene models fitting CNV-trait pairs For each CNV-trait pair, we specify whether pairwise models performed better than single gene models (all but one CNV-trait pair, left column), whether genes represented disproportionately in significant pairs primarily represented genes significant in a single gene model (half of pairs, middle column), and whether region-wide association with a trait was significant (BMI and IQ only, right column). Yes: blue, No: orange. Dotted fields in the first column represent cases where the pairwise model advantage did not exceed that of control regions. Dotted fields in the second column represent cases where they were no over-represented single genes in pairs.

that of CNV regions in 5 of the 7—was somewhat surprising due to our hypothesis that CNV regions have the unique property of dysregulation of nearby genes in the same direction. However, perhaps regulatory landscape across many regions of the genome also biases towards expression dysregulation of physically colocalized genes in the same direction, and CNV regions are particularly pathogenic due to specific pairs of genes or above-average density of associated pairs.

As we observed neither enrichment in the proportion of significant pairwise tests nor outlier top signal in the QQ plots, the pairwise contribution to explaining trait variance seems to be oligogenic across the region. However, in some cases we did observe outliers when examining the frequency of specific genes involved in top pairs. There was striking variation across traits and regions in terms of whether the top single genes were also the top contributors to pairs or novel genes were equally likely to contribute. A single gene was repeatedly contributing to top pairs for bipolar disorder at 16p11.2 (*INO80E*, 26% of top pairs) and schizophrenia at 22q11.2 (*PPIL2*, 42% of top pairs). The individual association with these genes was not detected, but the recurrent role of these genes in pairs suggests an important trait contribution. In contrast, for schizophrenia at 16p11.2 and ASD at 22q11.2, multiple top single genes participate disproportionately in top pairs. Intriguingly, although pairwise models show similar advantages for ASD at 16p11.2 and IQ at 22q11.2, genes across the region are more evenly represented in top pairs. Bipolar disorder at 22q11.2 (with single genes models most often explaining variance) showed association with flanking genes on either side, *TUBA8*, *TMEM191B*, and *PPIL2*; *PPIL2* appeared in most of the pairs, as well. Because we did not find overall support for a pairwise model for bipolar disorder at 22q11.2, this may simply reflect the independent association of *PPIL2*. Our finding of *PPIL2* as a bipolar disorder driving gene is supported by

this gene's over-representation of rare protein truncating variants in the Bipolar Exome sequencing consortium data [27]. However, it may also be notable that bipolar disorder has less evidence for association with these CNV regions than most of our other tested traits [28, 29].

Given that the pairwise signal tended to be oligogenic and that expression imputation of adjacent genes has high correlation (as well as additional limitations of our genetic architecture analysis discussed below), it is difficult to interpret association at the level of specific gene pairs. Despite low confidence in any single result, we find it potentially informative to review literature support and potential future directions for our most statistically convincing gene (pair) associations, particularly those that recur across pairs or phenotypes or are representative of a pattern between individual and pairwise top results. For ASD at 16p11.2, the top 15 pairs include four with *FAM57B*. This gene was previously shown to have multiple within-region interactions in zebrafish [23]. Here, we find that the top pairwise contributions are with coding and non-coding genes in repetitive or flanking regions (*RP11-347C12.3*, *TBC1D10B*, *BOLA2B*, *NPIPB12*). Studies of 16p11.2 CNV genes rarely include these flanking genes, but our data suggest that they may contribute to trait association. Notably, our expectation of expression dysregulation in the same direction would be less strong for flanking genes, so expanded testing of flanking regions may be worthwhile. The *FAM57B* and *DOC2A* pair, associated with hyperactivity, head size, and length in the zebrafish study, was in the top quarter of associated pairs for BMI and IQ. We note that McCammon *et al* specifically excluded additive effects, while our study is based on genes contributing additively to pairs (which we find explains more variance than traditional interactions). For BMI at 16p11.2, the top ranked pair is *CDIPT* with *ALDOA*. It is notable that these two genes were not top-ranked individual genes for BMI, demonstrating the utility of our pairwise approach to prioritize pairs that might not be detected as individual genes. The top pair for IQ, *MVP* and *KCTD13*, on the other hand, includes one top IQ-associated gene (*MVP*) and one gene (*KCTD13*) not associated with IQ. This finding is similar to an observation in zebrafish, where the expressivity of head-size phenotypes driven by *KCTD13* overexpression was increased by additional overexpression of *MVP* [13]. For IQ at 22q11.2, several top pairs contain *COMT* along with a non-coding gene. *COMT* is a gene with variants believed to affect multiple traits, including executive function and schizophrenia risk in the general population and executive function in 22q11.2 deletion carriers [30–33], and whose expression is associated with IQ in the general population [34]. Our data provides a refined hypothesis that the relationship between *COMT* and IQ is dependent on additional non-coding genes at 22q11.2. IQ is, in fact, the only trait for which coding-noncoding pairs are over-represented as significant hits with respect to coding-coding pairs, which may have implications for future genetic studies of IQ (S1, S2, S6 and S7 Tables).

We also wondered whether there was a general contribution across many genes in the region. In our analyses, we found that there was a region-wide contribution to both BMI and IQ in both CNVs. The large number of 16p11.2 genes associated with BMI in both single and pairwise models was consistent with a region-wide signal. From previously established associations in CNV carriers, we would expect a negative correlation between increased expression and BMI for both 16p11.2 and 22q11.2 CNVs. However, we saw this only at 22q11.2 in the region-wide model. Previously, we found individual genes independently associated with both increases and decreases of BMI at 16p11.2 [22]. We hypothesized that there may be both BMI-increasing and BMI-decreasing genes in the 16p11.2 region due to our observation of association in both directions in single-gene models (and BMI decreases in syntenic deletion mice [12]), in which case we might have been picking up more BMI-decreasing genes in our region-wide score. Alternatively, one known limitation of our cross-tissue expression prediction approach is that our results may not be driven by the biologically-relevant tissues and thus

appear to be opposite in direction [35]. We also note that BMI and IQ are quantitative traits with high sample size, and so we may have had power limitations in other traits.

Previously, we proposed that *INO80E* at 16p11.2 is a driver of schizophrenia and BMI, a finding that has been corroborated in similar analyses by others [22]. However, we found that pairwise models explained more trait variance in both schizophrenia and BMI at 16p11.2, so it is possible that the pathophysiological contribution of *INO80E* will be better explained in combination with other genes than independently, a hypothesis that might be of interest for experimental design. Our pairwise findings also suggest that *INO80E* has an important contribution to at least two other traits. In bipolar disorder, *INO80E* is the top individual associated gene and is the most disproportionate contributor to pairs. In ASD, *INO80E* is a weakly associated top individual gene and is the most frequent (albeit not strongly disproportionate) contributor to significant pairs. This finding suggests that four traits may be influenced by the *INO80E* gene, and at least in the case of ASD, this gene works in combination with other genes. However, we have not found evidence of the involvement of *INO80E* in IQ, highlighting that the neurobehavioral phenotypes of 16p11.2 may be broader than the impact of this single gene, under the assumption that IQ in the general population is a good representation of the 16p11.2-mediated impact on intellectual ability.

There are a number of limitations in our approach to probing the architecture of 16p11.2 and 22q11.2 CNVs using pairs of gene expression predictions and region-wide gene expression scores. There are numerous combinatorial models that have not been tested, and the true architecture of gene-trait pairs may lie anywhere in between what we can capture in simplified models. In fact, given the observation that the entire 16p chromosome arm is enriched for ASD risk signal and has high amount of chromosomal contact, the region itself, as we had defined it, could be insufficient [36–38]. By design, the expression imputation framework is based entirely on cis-regulation of expression and does not capture trans-regulatory effects. Thus an effect on trait that is due to non-cis architecture may be a false negative. Another plausible scenario is that a cis-eQTL has pleiotropy with a trans-regulatory region (such as chromosomal contact). If that is the case, within-region gene pairs that appear associated with phenotypes may be misleading with respect to specific genes, although our observations about the advantage of pairwise models for variance explained would remain valid. Finally, TWAS approaches are unable to capture direction of effect in a reliable way [39]. As a result, this experimental design also does not allow us to account for the interesting questions of directionality (i.e., in cases where a trait is present in only deletions or duplications) and reciprocal phenotypes.

Another potential model that we have not tested is that only the extremes of the distribution–either in pairwise sums or region-wide scores–will impact a phenotype, and more modest increases and decreases in gene expression are buffered. For example, the BMI-16p11.2 panel in Fig 3 suggests a difference in the top and bottom decile compared to the BMI-score relationship in the intermediate deciles. Our study using all individuals has an advantage in statistical power if more typical gene expression levels are relevant to the trait, but a disadvantage given the potential noise that is introduced if only extreme expression deviation is relevant to uncommon traits such as schizophrenia, bipolar disorder, and ASD.

A technical limitation of our study design is that available datasets are not always ideal for our approach. For BMI, IQ, and ASD, the best-powered datasets are summary statistics. We use the summary statistics for single and pairwise association testing, determining permutation-based significance cutoffs, and finding top individual genes that are represented in pairs. However, in order to measure variance explained and region-wide scoring, we use individual-level data. We have to consider heterogeneity across the cohorts as a caveat when comparing results. Still, for both ASD and IQ the individual level data used is a subset of the full cohort

comprising summary level statistics, minimizing the differences. For consistency with the published Psychiatric Genomics Consortium data, we limited our analyses to white (European) ancestry. As a result, we are likely not capturing the entire spectrum of eQTL data and GWAS and are less confident in the transferability of our expression prediction models, but we expect that our findings about CNV-trait architecture will be true for any genetic-ancestry group. Finally, our study is based on multiple tissues derived from adults, rather than more targeted analyses of the brain during specific developmental timepoints. Similarly, when we decide which model explains more variance, we do not weight tissues differently (according to trait relevance, sample size, etc.). Despite the limitations, we may be detecting signal driven by a subset of the data; for example, ASD-donor cerebral organoids show cell-type specificity of *INO80E* to neuroepithelial cells during development, yet we detect a pairwise contribution in cross-tissue analysis [40].

The 16p11.2 and 22q11.2 regions are highly penetrant for neurobehavioral traits, but require a better understanding of genetic architecture to indicate key biological pathways. By extending transcription imputation to study a simple summed model of pairwise gene expression, we uncover a consistent pattern of higher variance explained by gene pairs than either single genes or traditional interaction models and several traits showing region-wide association signal (Fig 4). Most of these patterns appear specific to CNV regions and did not appear to represent the genetic architecture in matched control regions. ASD, for which single gene approaches had small to no effect, shows pairwise association signal above that of controls at both 16p11.2 and 22q11.2. Having failed to dissect 22q11.2 with single-gene approaches, here we found least two 22q11.2 traits–BMI and IQ–that can be better modeled region-wide. Our study suggests that pathobiological insights might result from studying combinations of the genes in and near these CNVs, albeit with potentially differing genetic architecture across traits and regions.

## Methods

### Genes studied

We selected genes at the 16p11.2 and 22q11.2 CNV regions that fell into one of these annotation categories: protein-coding, lincRNA, pseudogene, antisense, miRNA. These were consistent with what was used for PrediXcan modeling previously, with miRNA included given the strong representation of miRNAs at 22q11.2 [41, 42]. We included noncoding genes, as they have not received significant attention in studies of these regions, despite some evidence of miRNA contribution to 22q11.2 phenotypes. In addition, we considered flanking genes within 200kb of the region, as there is suggestive evidence of broader transcriptional effects in CNV carriers, and because we previously found evidence of flanking gene involvement in psychosis [22, 27]. S1 and S2 Tables contain single and pairwise CNV genes used in analysis.

### Phenotypes and datasets

Imputed genotypes from the Psychiatric Genomics Consortium were used to study schizophrenia (wave 3 freeze), bipolar disorder (wave 2 freeze), and ASD (2019 freeze, used for analysis of variance explained only) [25, 43, 44]. Although bipolar disorder is not one of the strongest associated phenotypes with either CNV, it does get picked up in carrier screens, and we included it in the study due to its large sample size and high genetic overlap with schizophrenia [28, 29, 45, 46]. An additional joint PGC-iPsych ASD summary statistic set was used to boost power for ASD analyses (www.med.unc.edu/pgc/download-results/) [44]. Summary statistics from the GIANT consortium (2015 freeze, www.portals.broadinstitute.org/ collaboration/giant/index.php/GIANT_consortium_data_files)) were used to study BMI, and

a VU-Amsterdam University cohort (wave 2 freeze, www.ctg.cncr.nl/software/summary_statistics) was used for IQ [47, 48]. Individual-level IQ and BMI data from the UK Biobank were used for validating discoveries in individual-level data on phenotypes for which individual-level data were not available [49]. Cohorts and sample sizes are listed in S3 Table. It is possible that 1–2% of the individuals in these cohorts are CNV carriers, resulting in poorer expression prediction, but the amount of noise will be negligible and biased towards false negatives rather than false positives.

## Predicting the expression levels of individual 16p11.2 and 22q11.2 CNV genes

Analyses of single genes were performed using elastic net models from www.predictdb.org trained on the GTEx version 8 data [24]. These prediction models were available for up to 42 16p11.2 genes and up to 65 22q11.2 genes in at least one tissue. The elastic net approach was chosen for consistency with our pairwise model training approach. Gene expression for each CNV gene in each individual was predicted using the—predict option in PrediXcan, for each cohort [50]. Analyses on summary statistics did not require expression prediction.

## Finding control gene sets

To create control gene sets to use in a permutation-based analysis, the 16p11.2 and 22q11.2 regions were matched on three categories: (1) number of genes (exact), (2) length of the region (in bases, 80–120% of the region), (3) ratio of coding to non-coding genes (at least 80% that of the region to avoid picking up dense regions of noncoding genes). Gene sets that overlapped the distal 16p11.2 region or the Major Histocompatibility Complex (a known gene-dense major GWAS-identified locus for schizophrenia) were excluded [51]. Overall, we found 38 comparable regions to 16p11.2 and 28 to 22q11.2. The list of control regions can be found in S4 Table.

## Predicting the expression of pairs of 16p11.2 and 22q11.2 CNV genes

As a simple way to model pairwise gene expression, we took every pair of genes in each CNV and defined pairwise "joint genes" with expression equal to the inverse-normalized sum of the expressions of each gene in GTEx. Normalized GTEx gene expression sums were used as inputs to the PrediXcan elastic net model training pipeline (www.github.com/hakyimlab/PredictDB_Pipeline_GTEx_v7), with covariates used for the GTEx v8 analyses downloaded from www.gtexportal.org/home/datasets. Given that our goal was to evaluate the contribution of these pairwise genes to specific traits, rather than a general-use pairwise model training process, a high overlap between the SNPs in our models and the GWAS datasets was vital. For that reason, we chose to repeat the training process for each trait, leaving only the SNPs in each GWAS dataset as inputs for model training. We repeated this model training process again on the control pairs of genes. For 16p11.2 genes, the prediction $R^2$ median and mean are 0.091 and 0.12, respectively; for chosen controls they are 0.093 and 0.14. For 22q11.2 genes, the prediction median and mean are 0.1 and 0.13, respectively; for chosen controls they are 0.09 and 0.13. This consistency suggests that the control regions are well-matched for key properties relevant to predicted expression.

To generate the plots in S1 Fig, we extracted predictive SNPs for the *INO80E/ZNF48* gene pair in the Frontal Cortex-BA9 tissue, as well as the SNPs for the two individual genes. Locus-Zoom (original version, www.locuszoom.org) was run using the schizophrenia association results for these SNPs from the PGC Schizophrenia wave 3 summary statistics (https://doi.org/10.6084/m9.figshare.19426775). LD correlation values were generated by LDMatrix from

LDLink (www.ldlink.nci.nih.gov) and plotted using the LDHeatmap R package with a color key intended to match that of LocuzZoom.

## Association studies between predicted expression and traits

**Individual level.**   Each PGC cohort was converted from PLINK to dosage format for PrediXcan input. Tissue-specific prediction models were applied to each tissue in each cohort. MultiXcan, a cross-tissue implementation of PrediXcan, was used to combine predicted expressions across tissues and perform association with trait [52]. Using the MultiXcan p-value and the average direction of effect of each gene across tissues, we used METAL to determine a per-gene result [53]. Both single gene and pairwise analyses were performed in the same way.

**Summary level.**   The 'MetaMany' option in the MetaXcan package was applied to summary-level data using single-tissue prediction models to generate gene-level association results for each tissue [54]. S-MultiXcan, a cross-tissue implementation of PrediXcan for summary level data, was used to combine across tissues for tissue-wide association results [52]. Cross-tissue covariances were downloaded from PredictDB for single-gene models and generated from single tissue covariances for pairwise models using the covariance builder script available at www.github.com/hakyimlab/MetaXcan/blob/master/software/CovarianceBuilder.py. Both single gene and pairwise analyses of summary statistics were performed in the same way.

## UK Biobank additional expression prediction

While the best-powered GWAS meta-analyses of BMI and IQ were available as summary statistics, certain analyses such as interaction models and percent variance explained require individual-level data. We obtained IQ and BMI measurements from the UK Biobank and took an average across visits for people with multiple measurements. Analysis was limited to individuals who reported their ethnicity as "white", and included age, age-squared, and 40 principal components as covariates. A large number of principal components was used due to the correlation between the BMI phenotype and components in the PC 30–40 range. Expression imputation for single genes and pairs was performed with PrediXcan as described above.

## Significance thresholds for association studies

For all association studies, a permutation-based threshold was determined using the control gene sets. After association testing between control gene sets for a CNV and phenotype, the median of the 5th percentile of control sets was used as a 5% significance threshold for the true CNV region. As control genes are chosen independently of association with trait, using a median across all regions will counteract bias caused by any control gene set overlapping a strong GWAS peak for a trait.

## Estimating variance explained by pairwise models

Variance in phenotype explained by imputed expression was measured as the $R^2$ of the linear model between case-control status and imputed expression for all genes in the CNV. Specifically, the adjusted $R^2$ was used, as using all pairs of genes involves a large number of variables. For every tissue-cohort pair, $R^2$ values were calculated using all single genes, all pairwise genes, and interaction terms. The number of times a model (single, pairwise, or interaction) had the greatest $R^2$ for a cohort-tissue pair was tallied. The same process was implemented for control gene sets. A chi-square test was performed to determine whether the proportion of pairwise models being "best" in a CNV region was different from the proportion in control regions.

This approach required individual-level data, and as we used summary level data for ASD, IQ, and BMI, we performed it in PGC ASD individual-level data (without iPSYCH), and UK Biobank for IQ and BMI (each of which was treated as one single cohort).

We acknowledge that previous attempts to solve the problem of variance explained by predicted expression were made by Liang et al [55]. We attempted this method and found extremely large estimates for variance explained. This inflation might be due to our relatively small (<5 MB) regions of interest with high SNP and predicted-expression correlation structure, as opposed to a predicted transcriptome-wide screen. The estimates provided by our approach, where the adjusted $R^2$ rarely exceeds 0.05, are a more reasonable estimate of the effect of one small set of genes on a trait.

## Testing a region-wide model

We estimated a region-wide score for each individual using their single-gene predicted expressions. First, we found the normalized rank of an individual for the expression of a gene; the median rank was used for genes expressed in multiple tissues. The sum of an individual's gene-specific rankings became the individual's region-wide score; these scores were converted to normalized (between 0 and 1) ranks. For quantitative traits, we quantified the relationship between score and phenotype as a Pearson correlation. For binary traits, we tested for a difference in score distribution between cases and controls (Kolmogorov-Smirnov test), as well as for a difference in score means between cases and controls (t-test).

Initially, we attempted to approach region-wide association in the same way as pairwise association for schizophrenia. Region-wide sums of GTEx expressions of all CNV genes were used as inputs into elastic net models from GTEx SNPs, with the same covariates as before. After model quality filtering, models in only 5 tissues at 16p11.2 and 13 tissues at 22q11.2 remained, all with $R^2 < 0.1$. As a result, we did not further pursue this method.

## Supporting information

**S1 Fig. Landscape of SNP predictors for single and pairwise genes.** As a representative example of the relationship between the prediction SNP architecture for single vs. pairwise genes, we selected *INO80E* and *ZNF48*, the top schizophrenia pair, in the frontal cortex, a putative schizophrenia-relevant tissue. 60 SNPs from three categories are plotted: 20 *INO80E* predictors, 32 *ZNF48* predictors, and 8 predictors of the *INO80E/ZNF48* pair. Top: LocusZoom plot of association of predictor SNPs with schizophrenia. Circles: SNP predictors in one of the three (above) categories. Y-axis: schizophrenia association *P*-value (left, circles); recombination rate (right, blue line). X-axis: distance on chr. 16. Middle: Distribution of plotted SNP positions for all three categories (vertical bars) with colored lines representing prediction for gene or pair. Red rectangle: *INO80E* gene, red lines: *INO80E* predictors; blue rectangle: *ZNF48* gene, blue lines: *ZNF48* predictors; purple x: *INO80E-ZNF48* theoretical expression pair, purple lines, *INO80E-ZNF48* pair predictors. Bottom: LD structure of predictive SNPs. LD heatmap color scale is in the same order as the $R^2$ scale in LocusZoom. Note: Approximately 120 additional unique genes (both coding and noncoding) are located in the region (not shown), including 30 between *INO80E* and *ZNF48*.
(PDF)

**S2 Fig. Pairwise signal at 16p11.2 is polygenic.** Q-Q plots comparing PrediXcan association signal from single 16p11.2 genes (blue), pairs of 16p11.2 genes (orange), single genes in control subsets (gray), and pairs of genes in control subsets (black).
(PDF)

**S3 Fig. Pairwise signal at 22q11.2 is polygenic.** Q-Q plots comparing PrediXcan association signal from single 22q11.2 genes (blue), pairs of 22q11.2 genes (orange), single genes in control subsets (gray), and pairs of genes in control subsets (black).
(PDF)

**S4 Fig. Patterns of top single genes contributing to significant pairs at 16p11.2.** Y-axis: counts of the number of times each gene contributes to a significant pair (permutation *P*-value < median of 5th percentiles of control region p-values). Bars in blue represent genes significant (permutation *P*-value < median of 5th percentiles of control region p-values) in a single gene model for the same trait, with rank indicated above the bar. Bars in orange represent genes not significant in a single gene model. X-axis: genes in chromosomal order. Disproportionately represented genes (mean + 2.5 standard deviations) are bolded.
(PDF)

**S5 Fig. Patterns of top single genes contributing to significant pairs at 22q11.2.** Y-axis: counts of the number of times each gene contributes to a significant pair (permutation *P*-value < median of 5th percentiles of control region p-values). Bars in blue represent genes significant (permutation *P*-value < median of 5th percentiles of control region p-values) in a single gene model for the same trait, with rank indicated above the bar. Bars in orange represent genes not significant in a single gene model. X-axis: genes in chromosomal order. Disproportionately represented genes (mean + 2.5 standard deviations) are bolded.
(PDF)

**S6 Fig. Region-wide score association with ASD, bipolar disorder, and schizophrenia.** Region-wide scores across individuals were binned into deciles and the mean (dot) and standard error (bars) of case-control ratios for each decile are plotted. Best fit line across deciles is shown. Top: 16p11.2. Bottom: 22q11.2. Left to right: ASD, Schizophrenia, Bipolar Disorder.
(PDF)

**S1 Note. Members of the Psychiatric Genomics Consortium contributing to this work.**
(DOCX)

**S1 Table. Pairwise and single predictive model qualities for 16p11.2 genes.** Four different model sets were created for SNP overlap (ASD and bipolar used the same PGC panel). For each pair of genes, the median of prediction qualities ($R^2$) among tissues along with the number of tissues for which predictive models are available (which can be zero) are noted. These are compared to single-gene predictive model qualities and coding/noncoding genes are annotated.
(XLSX)

**S2 Table. Pairwise and single predictive model qualities for 22q11.2 genes.** Four different model sets were created for SNP overlap (ASD and bipolar used the same PGC panel). For each pair of genes, the median of prediction qualities ($R^2$) among tissues along with the number of tissues for which predictive models are available (which can be zero) are noted. These are compared to single-gene predictive model qualities and coding/noncoding genes are annotated.
(XLSX)

**S3 Table. Cohorts used for analyses.** The specific cohorts from the Psychiatric Genomics Consortium that were used for this analysis are listed.
(XLSX)

**S4 Table. Genomic regions used as controls for analyses.** Each region is annotated with genomic context (for example if it is dense with olfactory genes). Regions that were matched to each CNV but were not used for analysis due to MHC or study CNV overlap are labeled as such.
(XLSX)

**S5 Table. Single gene associations with five neurobehavioral traits.** Genes are listed in order of MultiXcan/S-MultiXcan p-values and genes significant based on a permutation-based threshold are highlighted.
(XLSX)

**S6 Table. Pairwise associations with five neurobehavioral traits at 16p11.2.** Genes are listed in order of MultiXcan/S-MultiXcan p-values and genes significant based on a permutation-based threshold are highlighted.
(XLSX)

**S7 Table. Pairwise associations with five neurobehavioral traits at 22q11.2.** Genes are listed in order of MultiXcan/S-MultiXcan p-values and genes significant based on a permutation-based threshold are highlighted.
(XLSX)

**S8 Table. Number of times a 16p11.2 or 22q11.2 gene is represented in a top associated pair.** Tabulation of count data from Fig 2. Highlighted genes are significant single genes for each trait. Genes that are starred are disproportionately represented (count > mean + 2.5 SD).
(XLSX)

## Acknowledgments

We acknowledge Nancy J. Cox for her contribution to study conception and advice about methodology. Noah Zaitlen provided helpful ideas for testing the utility of pairwise models. This research has been conducted using the UK Biobank Resource under Application Number 47982.

## Author Contributions

**Conceptualization:** Mikhail Vysotskiy, Lauren A. Weiss.

**Data curation:** Mikhail Vysotskiy.

**Formal analysis:** Mikhail Vysotskiy.

**Funding acquisition:** Lauren A. Weiss.

**Investigation:** Mikhail Vysotskiy.

**Methodology:** Mikhail Vysotskiy, Lauren A. Weiss.

**Project administration:** Lauren A. Weiss.

**Resources:** Mikhail Vysotskiy, Lauren A. Weiss.

**Software:** Mikhail Vysotskiy.

**Supervision:** Lauren A. Weiss.

**Validation:** Mikhail Vysotskiy.

**Visualization:** Mikhail Vysotskiy.

**Writing – original draft:** Mikhail Vysotskiy, Lauren A. Weiss.

**Writing – review & editing:** Mikhail Vysotskiy, Lauren A. Weiss.

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
