## [Decision Letter · Decision Letter 0]

22 Nov 2022

Dear Dr Vysotskiy,

Thank you very much for submitting your Research Article entitled 'Combinations of genes at the 16p11.2 and 22q11.2 CNVs contribute to neurobehavioral traits' to PLOS Genetics. I apologise for the delay in getting the reviews back to you.

The manuscript was discussed in detail at the editorial level and fully evaluated by independent peer reviewers. The reviewers appreciated the attention to an important problem, but raised some substantial concerns about the current manuscript, including issues regarding the clarity of methods used, role of regulatory regions within the CNV regions tested, and choice of phenotypes assessed in this study.

Based on the reviews, we will not be able to accept this version of the manuscript, but we would be willing to review a much-revised version. We cannot, of course, promise publication at that time.

If you decide to revise the manuscript for further consideration at PLOS Genetics, please aim to resubmit within the next 60 days, unless it will take extra time to address the concerns of the reviewers, in which case we would appreciate an expected resubmission date by email to plosgenetics@plos.org.

We are sorry that we cannot be more positive about your manuscript at this stage. Please do not hesitate to contact us if you have any concerns or questions.

Yours sincerely,

Santhosh Girirajan

Academic Editor

PLOS Genetics

Hua Tang

Section Editor

PLOS Genetics

Reviewer's Responses to Questions

**Comments to the Authors:**

Reviewer #1: This manuscript takes an interesting approach to investigating 16p11.2 and 22q11.2 copy number variants (CNVs); these CNVs are strongly associated with multiple neurobehavioral phenotypes. As each of these loci contain multiple genes, identifying specific genes contributing to each disorder/phenotype and dissecting the architecture of CNV-trait associations has been challenging. This manuscript uses expression-imputation methodology to generate pairwise expression imputation models for CNV genes and then applies the models to GWAS for several neurobehavioral traits. They compared the trait variance explained by pairs of genes expressed in the same direction with the variance explained with single genes and with traditional interaction models, in addition to modeling polygenic region-wide effects. The authors find that for all CNV-trait pairs -except for bipolar disorder at 22q11.2- pairwise effects explain more variance than single genes. Individual genes over-represented in top pairs did not show single-gene signal. They also found, testing for region-wide association, that both loci have significant effects on BMI and IQ . Notably, differences in genetic architecture are found by trait and region, but the vast majority (9/10) CNV-trait combinations showed evidence for multigene contribution; for most, the importance of combinatorial models appeared unique to CNV regions.

The authors have recently used this approach to test whether individual genes in the 16p11.2 and 22q11.2 CNV regions were contributing to the same five traits of interest investigated here, and found a couple of 16p11.2 gene associations but none for 22q11.2 (Vysotskiy et al, Genome Med 2021) , motivating their current investigation of combinatorial associations in these traits of interest.

The manuscript addresses an important and highly relevant topic, and is clearly written. As the authors note, using eQTLs and expression prediction is likely to find additional information missed by GWAS analyses, so this is an interesting approach. The comparison to control gene sets matched on 3 categories is also a strength. The conclusion that ‘mechanistic insights for CNV pathology may require combinational models’ is not surprising, but it is nice to see this demonstrated empirically in a new way.

I do have some questions about the premise and the approach, however. While the pairwise approach certainly makes more sense than investigation of single genes for these CNVs that elevate risk for highly polygenic traits, the expression-imputation methodology utilized here also rests on the critical assumption that genetic regulation is similar in cases and controls, which may not be the case. Further, the approach does not consider the vast regulatory landscape of both of these CNVs (e.g., https://www.medrxiv.org/content/10.1101/2022.03.23.22272826v1). 22q11.2 and 16p11.2 are genomic regions with the densest intrachromosomal contact; indeed 22q11.2 has the highest density of enhancer elements of any similarly sized genomic region. As such, broader disruptions of regulatory architecture may be explaining more of the variance in these neurobehavioral phenotypes than specific gene pairs in the loci. The authors state in the Discussion that ‘perhaps [the] regulatory landscape across many regions of the genome also biases towards expression dysregulation of physically colocalized genes in the same direction’ ; which may be correct , but seems to go against their hypothesis; and then would also not explain why these genomic regions have such a uniquely strong contribution to neurobehavioral phenotypes.

On this point, I also am curious about their choice of these 5 phenotypes. There is no compelling association of bipolar disorder with these CNVs (certainly associations with other developmental neuropsychiatric disorders- ADHD, anxiety disorder- are stronger), and while there is one small study reporting on obesity in adults with 22q11.2 deletions this is a quite different level of evidence compared to the robust opposing anthropometric phenotypes observed for 16p11.2 reciprocal CNVs (e.g , Jacquemont et al, and other papers cited). Further, at least for BMI in 16p11.2 and for schizophrenia (possibly for both loci) there is evidence of opposing effects of the deletion vs duplication, but similar direction of effect for ASD and IQ. This doesn’t appear to be directly addressed in the manuscript.

Given the lack of evidence for association of bipolar disorder with these CNVs, it is unclear how to interpret the distinct pattern for bipolar disorder compared to the other traits- ie, that at both loci, one gene (INO80E and PPIL2 respectively) stood out as a disproportionate contributor to pairs. Could the authors comment on this in the Discussion?

Given that region-wide associations were only observed for the quantitative traits analyzed, is there any concern this could be a psychometric issue?

Also – was there any consistency or pattern to the types of genes (protein-coding, lincRNA, pseudogene, antisense, miRNA) showing strongest pairwise signal, across the 2 loci?

Analyses were limited to those of ‘white’ ancestry – while perhaps this couldn’t be avoided, it certainly needs to be directly discussed as a limitation.

Reviewer #2: Using TWAS implemented in PrediXcan the authors attempt to understand the contribution of individual genes within two described microdeletion/duplication syndromes. The authors selected the genes encompassed by the syndromes for analysis (not specified), used summary statistics from the Psychiatric Genomics and GIANT consortia for various neuropsychiatric traits and conditions, and used predicted transcription levels from the updated GTEx v8 release. 38 control regions were selected for permutation-based analyses by number of genes, length, ratio of coding to non-coding genes, notably GC content was not accounted for.

The work is interesting and attempts to answer longstanding questions of interest for microdeletion/duplication syndromes, however the descriptions of the methodological are insufficient for me to clearly understand the novelty of the approach and a more clear explanation and presentation of the findings could be improved.

General comments:

The abstract does not outline the methodological approaches with much clarity and does not highlight specific results of interest. Some comment about the patient selection in the GWAS consortia is needed—it is likely that individuals with 22q11.2 deletion syndrome were included in these cohorts as the disorders are fairly common and underdiagnosed when congenital heart disease or thymic agenesis are not prominent phenotypes.

The first sentence of the results should describe in a bit more detail the experimental approach used by the authors. Even after reading the methods I was still somewhat confused about the approach—was PrediXcan performed for each deletion/duplication gene along with the ‘control genes’ and the p-value derived from the permutation of these test-statistics? Was the significance threshold for the findings pre-specified? How were the pairwise signals calculated? An explanatory figure on the analytical approach and permutation procedure might be necessary for the general reader.

Figure 2:

Gene names are difficult to read and great in number. Would something like a regional Manhattan plot be more legible with genes annotated like the genome browser? The Y-axis “appearance in top pairs” – is not a familiar statistic to me. P-values with a clear explanation of their derivation from the permutation test might make more sense. A small note on style—I would encourage the authors to use something other than default ggplot colors.

Figure 3:

Further explanation is needed. The figure title notes an association but I do not see a statistical description of association or a p-value that would clearly indicate this interpretation. What is a region-wide score decile?

Figure 4:

The title “Insights gained into CNV-trait pairs” should be more descriptive—what are the insights specifically?

**Have all data underlying the figures and results presented in the manuscript been provided?**

Reviewer #1: Yes

Reviewer #2: Yes

PLOS authors have the option to publish the peer review history of their article (what does this mean?). If published, this will include your full peer review and any attached files.

Reviewer #1: No

Reviewer #2: No

---

## [Decision Letter · Decision Letter 1]

31 Mar 2023

Dear Dr Vysotskiy,

Thank you very much for submitting your Research Article entitled 'Combinations of genes at the 16p11.2 and 22q11.2 CNVs contribute to neurobehavioral traits' to PLOS Genetics.

The manuscript was fully evaluated at the editorial level and sent for review again to peer reviewers. The reviewers noted that their major comments were addressed but identified some concerns that might help improve the clarity of the presented work. We therefore request you to modify the manuscript according to the review recommendations. Your revisions should address the specific points made by each reviewer.

Yours sincerely,

Santhosh Girirajan

Academic Editor

PLOS Genetics

Hua Tang

Section Editor

PLOS Genetics

Reviewer's Responses to Questions

**Comments to the Authors:**

Reviewer #1: Overall, the authors’ hypothesis in this manuscript is that multiple genes acting together within a CNV region (particularly the 1611.2 and 22q11.2 CNVs) have greater explanatory power for CNV association with a trait than does a single gene in a CNV region. The authors have been responsive to the concerns for the prior review, and have provided more clarification regarding their analysis framework.

One of my major concerns with the original manuscript was potential for false negatives with the approach, in the case of chromosomal contact and enhancer mediated effects that play a role in CNV pathogenicity. As the authors acknowledge, this concern can’t be addressed with their study design; however, they have provided a much more thorough consideration of the issue (and several others raised in the review) in the limitations, which I appreciate.

One remaining concern I have is the focus on the specific gene pairs in the discussion may be premature, given that -in the event of pleiotropy with trans-regulatory regions- the within-region gene pairs that appear associated with phenotypes ‘may be misleading with respect to specific genes’ (as the authors state in their response). We agree that the observations about the advantage of pairwise models for variance explained would still remain valid, but it would call into serious question any conclusions about the specific genes. Without further information about how likely this scenario is, I still think the conclusions about the specific gene pairs identified by this approach ought to be further tempered.

With regard to the effect of COMT on IQ in the context of 22q11.2 deletions , in the discussion the authors may also wish to cite Bearden et al AJP ( https://ajp.psychiatryonline.org/doi/full/10.1176/appi.ajp.161.9.1700)

Reviewer #2: The authors have responded to reviewer comments from myself and other reviewers which have resulted in improvements to the paper. I appreciate that there is a new figure which attempts to explain the overall schema of the analysis (as a side note I think that using a non-linear method like decision trees could be a fruitful approach to detecting interactions and non-additive effects between 2 or more genes on gene expression -- which would be outside the scope of this paper).

As I noted in the previous review, the representation of the data Figure 2 as essentially a large bar chart sorted in the order of the genomic locus is a missed opportunity. The data could either be more clearly conveyed in a table which would at least be more readable, or as a modified LocusZoom plot which would then show the relative length of each gene and give readers the opportunity to visualize how genes might be physically related to known cis-regulatory regions.

The information in Figure 3 also remains somewhat abstruse--this may be the most powerful set of information in the whole manuscript demonstrating (as I am understanding) that the predicted region-wide gene expression scores are related to some key continuous traits. I am honestly unsure how to convey the information more clearly--perhaps with a trendline relating the two variables for each region and an accompanying p-value?

**Have all data underlying the figures and results presented in the manuscript been provided?**

Reviewer #1: Yes

Reviewer #2: Yes

PLOS authors have the option to publish the peer review history of their article (what does this mean?). If published, this will include your full peer review and any attached files.

Reviewer #1: No

Reviewer #2: No

---

## [Editor Report · Decision Letter 2]

9 May 2023

Dear Dr Vysotskiy,

We are pleased to inform you that your manuscript entitled "Combinations of genes at the 16p11.2 and 22q11.2 CNVs contribute to neurobehavioral traits" has been editorially accepted for publication in PLOS Genetics. Congratulations!

Yours sincerely,

Santhosh Girirajan

Academic Editor

PLOS Genetics

Hua Tang

Section Editor

PLOS Genetics

Comments from the reviewers (if applicable):

**Data Deposition**

http://datadryad.org/submit?journalID=pgenetics&manu=PGENETICS-D-22-01128R2

**Press Queries**

---

## [Editor Report · Acceptance letter]

24 May 2023

PGENETICS-D-22-01128R2 

Combinations of genes at the 16p11.2 and 22q11.2 CNVs contribute to neurobehavioral traits 

Dear Dr Vysotskiy, 

We are pleased to inform you that your manuscript entitled "Combinations of genes at the 16p11.2 and 22q11.2 CNVs contribute to neurobehavioral traits" has been formally accepted for publication in PLOS Genetics! Your manuscript is now with our production department and you will be notified of the publication date in due course.

With kind regards,

Zsofi Zombor

PLOS Genetics

On behalf of:
